# Dirt Track Surface Preparation and Associated Differences in Speed, Stride Length, and Stride Frequency in Galloping Horses

**DOI:** 10.3390/s24082441

**Published:** 2024-04-11

**Authors:** Thilo Pfau, Olivia L. Bruce, Andrew Sawatsky, Renaud Leguillette, W. Brent Edwards

**Affiliations:** 1Faculty of Kinesiology, University of Calgary, Calgary, AB T2N 1N4, Canada; ajsawats@ucalgary.ca (A.S.); wbedward@ucalgary.ca (W.B.E.); 2Faculty of Veterinary Medicine, University of Calgary, Calgary, AB T2N 1N4, Canada; rleguill@ucalgary.ca; 3Department of Biomedical Engineering, Schulich School of Engineering, University of Calgary, Calgary, AB T2N 1N4, Canada; obruce@stanford.edu; 4McCaig Institute for Bone and Joint Health, Cumming School of Medicine, University of Calgary, Calgary, AB T2N 1N4, Canada

**Keywords:** horse, gallop, global positioning system, speed, stride length, stride frequency, track properties

## Abstract

In racehorses, the risk of musculoskeletal injury is linked to a decrease in speed and stride length (SL) over consecutive races prior to injury. Surface characteristics influence stride parameters. We hypothesized that large changes in stride parameters are found during galloping in response to dirt racetrack preparation. Harrowing of the back stretch of a half-mile dirt racetrack was altered in three individual lanes with decreasing depth from the inside to the outside. Track underlay compaction and water content were changed between days. Twelve horses (six on day 2) were sequentially galloped at a target speed of 16 ms^−1^ across the three lanes. Speed, stride frequency (SF), and SL were quantified with a GPS/GNSS logger. Mixed linear models with speed as covariate analyzed SF and SL, with track hardness and moisture content as fixed factors (*p* < 0.05). At the average speed of 16.48 ms^−1^, hardness (both *p* < 0.001) and moisture content (both *p* < 0.001) had significant effects on SF and SL. The largest difference in SL of 0.186 m between hardness and moisture conditions exceeded the 0.10 m longitudinal decrease over consecutive race starts previously identified as injury predictor. This suggests that detailed measurements of track conditions might be useful for refining injury prediction models.

## 1. Introduction

In racehorses, an increased risk of musculoskeletal injury has been linked to a decrease in speed and stride length over consecutive races prior to injury [1]. Surface type and track condition also influence stride parameters [1,2]. Environmental conditions modify surface properties [3,4]. Harder surfaces generally lead to increased impact shock, higher maximum ground reaction force, increased rate of force development [5,6] and a shortened slide phase during landing [7]. Soft ground has been speculated to lead to increased fatigue [2], which in turn, has been identified as a risk factor for injuries, for example, in eventing [8].

There are an increasing number of devices suitable for “in-training” and “in-race” quantification of stride parameters available for horses. Establishing the validity of these devices appears important [9,10]. This is a prerequisite of using such devices for the provision of more detailed information about confounding factors for improving existing injury prediction models [9]. Due to the outlined fundamental relationships between racing surface and stride parameters, one area of interest is the type of track surface and specific track conditions paralleling efforts in other equestrian disciplines [4]. This may provide higher specificity for the differentiation of injury-related changes and surface-associated effects, for example, from start to finish in a race or training session or across different lanes of a track.

In legged locomotion, speed is the product of stride length and stride frequency. Hence, an increase in speed either necessitates an increase in stride length, i.e., each individual stride covering a longer distance, or an increase in stride frequency, i.e., taking a higher number of strides per time unit, or a combination of both. At low to moderate speeds, horses in walk appear to increase stride frequency logarithmically with speed and linearly with speed in trot, while stride frequency is near constant in canter [11]. At higher canter and gallop speeds, stride frequency appears to increase more linearly with speed [12]. There are also reported training effects with time over the season [13]. Two-year-olds showed an initial decrease in stride duration with time that flattened off towards the end of the season, while only three-year-olds that had been given time off showed a decrease in stride duration [13]. There are also changes in stride parameters when galloping around bends, with higher duty factors recorded for the inside leg and a decrease in duty factor with increasing centripetal acceleration [14]. For artificial and turf surfaces, surface type and subjectively rated track condition through stewards’ assessment have been shown to influence race speed and stride length [1].

With the availability of small devices suitable for use “in-training” or “in-race”, additional aspects of the complex interaction between speed, stride frequency, and stride length can now be studied under “real life” conditions and in large sample sizes. Our study aimed at investigating a specific aspect associated with the horse–surface interaction. We were interested in documenting the presence and extent of changes in speed and stride length, as proven longitudinal between race injury predictors [1] in horses galloping on a dirt racing surface in relation to altering track preparation. Specifically, that study showed that for each decrease in speed of 0.1 ms^−1^ the injury risk increases by a factor of 1.18 over time (consecutive career race starts), and for each decrease in stride length of 10 cm, there is an increase in injury risk by a factor of 1.11 [1]. We hypothesized that, with changes in track hardness and moisture content, stride parameters and speed vary in a predictable manner and that, at a standardized speed of 16 ms^−1^, the observed changes exceed the longitudinal between race reduction in stride length over consecutive career race starts that have been identified as an injury predictor [1].

## 2. Materials and Methods

### 2.1. Track Preparation

A straight-line section of a dirt racetrack was separated into three lanes, each prepared differently by altering the depth of the surface cushion (top layer) with decreasing depth from the inside lane to the outside lane. On day 1, a less compacted (softer) track underlay was used, and on day 2, a more compacted track underlay and lower harrowing depths aiming at harder track conditions were used. On day 1, combinations of roller weights, the amount of water added to the track, and harrowing depths were used by the racetrack preparation crew to prepare the racetrack in accordance with the procedures used during the Calgary Stampede Chuckwagon races (day 1). On day 2, the crew was instructed to prepare the track with characteristics aiming at achieving faster racing times. No measurements of harrowing depths or amount of water used were undertaken.

### 2.2. Horses

Twelve Thoroughbred horses (all geldings, aged 8–15 years, mass 494.7 ± 30.7 kg) in training as Chuckwagon outrider horses were equipped with 10 Hz GPS-loggers (Vbox Sport, Racelogic, Novi, MI, USA) attached to the saddle pad caudal to the saddle. Nine horses were shod with aluminum race plates, one horse was shod with steel race plates, and two horses were barefoot. All horses were deemed “fit to compete” by their trainers and showed no obvious signs of gait abnormalities. Procedures were approved by the University of Calgary Animal Care Committee (AC21-0231), and written consent was obtained prior to testing by the owner.

### 2.3. Experimental Protocol

#### 2.3.1. Horse Exercise Protocol

Each horse was ridden in gallop by one experienced jockey (the same for all horses; male 73.5 kg body mass) on a 500 m section of a half-mile racetrack (Calgary Stampede, Calgary, AB, Canada) to a target speed of 16 ms^−1^ guided by a GPS watch on the jockey’s wrist (fenix 6, Garmin, Cochrane, AB, Canada). Horses repeated three runs per day, one for each lane in randomized order. After the first session (session 1) with the first four horses, the track was prepared for the second session in an identical fashion to prior to data collection. In the second session (session 2), the remining eight horses conducted their three runs. On the second day, six of the twelve horses performed an additional three runs with the track prepared with increased compression of the track underlay (session 3).

#### 2.3.2. Hardness and Moisture Measurements

Track hardness was measured according to ASTM testing standards F355 and F1936 with an impact testing device consisting of a uniaxial accelerometer (352C22 ICP accelerometer, ±500 g range, PCB Piezotronics, Depew, NY, USA) attached to a 9.1 kg mass that was dropped from a height of 65.5 cm through a guidance tube onto the surface [15]. Three locations along each lane (marked with pylons) were used for measurements, and recordings were undertaken at a sample rate of 1000 Hz with a WinDAQ analogue digital converter (DataQ Instruments, Akron, OH, USA). Three repeat measurements of moisture content were undertaken at the same sites that had been used for hardness measurements using a soil moisture meter (MO750, Extech Instruments, Waltham, MA, USA). Both hardness and moisture measurements were undertaken in close temporal proximity to the horse exercises, i.e., either directly before or after each of the three data collection sessions.

### 2.4. Data Processing

#### 2.4.1. Speed, Stride Frequency, and Stride Length

The calculation of stride frequency and stride length from the GPS/GNSS data followed published protocols [10]. In summary, GPS/GNSS speed spectrograms were created based on 64-sample Hamming windows, shifted 1 sample at a time over a 6 s portion of data applying a 1024-point FFT to each window. The frequency band (between 2 and 3 Hz) containing the maximum signal power was determined. Stride length (in m) was calculated from the associated speed value (in ms^−1^) at the timepoint of the associated Hamming window used for determining stride frequency. Stride length was calculated as speed divided by stride frequency.

#### 2.4.2. Track Hardness

Acceleration data were converted from voltage (V) to acceleration expressed in multiples of gravitational acceleration (g). For each recorded surface tester impact, the peak was identified (manually, custom software written in MATLAB), and the average of the three sites was calculated for each lane of each session. Hardness categories for statistical modelling were defined as follows: “soft” for all values ≤ 26 times gravitational acceleration, “medium” for values between >26 and <61 times gravitational acceleration, and “hard” for values ≥ 61 times gravitational acceleration.

#### 2.4.3. Moisture Content

Moisture categories (for statistical modeling) were based on the average readings for each “lane” of each session as follows: “dry”, moisture content ≤ 13%; “medium”, moisture content > 13 and ≤19%; and “wet”, moisture content > 19%.

### 2.5. Statistics

#### 2.5.1. Effect of Hardness and Moisture Content

Mixed linear models for stride frequency and stride length as outcome variables were implemented (SPSS v29, IBM, Armonk, NY, USA) with horse as a random factor and fixed factors, including speed as a continuous covariate, and surface characteristics (“hardness”, “moisture”) and shoeing condition (“aluminum”, “steel”, “barefoot”) as categorical fixed factors. Two-way interactions of fixed factors were also included in the initial model. After running initial models, including all two-way interactions, two-way interactions with *p*-values > 0.1 were eliminated, and the final multivariable model was created. Bonferroni post hoc tests were used, where applicable, for pairwise comparisons at a corrected level of *p* < 0.05.

#### 2.5.2. Effect of “Session” and “Lane”

A second set of mixed linear models for stride frequency and stride length as outcome parameters were run. Instead of using categorical track hardness categories and moisture levels as fixed factors, session (1: day 1 morning; 2: day 1 afternoon; 3: day 2 morning), and lane (inside, middle, outside) were used as fixed factors. Horse was used as a random factor, speed was used as a fixed continuous covariate, and shoeing, session, and lane and their two-way interactions were used as fixed factors. This model was implemented to evaluate whether the chosen surface characteristics, hardness and moisture content, are likely sufficient for explaining changes in stride parameters as a function of speed on the chosen dirt racing surface. Should more pronounced differences in stride parameters be found between the individual “session” × “lane” combinations compared to the “hardness” × “moisture” combinations, further track surface properties may be needed to explain these. A Bonferroni post hoc test was also applied to pairwise comparisons of these models at a corrected level of *p* < 0.05.

## 3. Results

### 3.1. Hardness and Moisture Content

Track hardness measurements across the two experimental days ranged from 22.33 to 61.19 times gravitational peak acceleration, and moisture content ranged from 12.14% to 27.38%. See Table 1 for the values measured for each lane for each session and the associated categorization used for statistical modeling.

### 3.2. Speed, Stride Frequency, and Stride Length

All GPS/GNSS recordings provided valid data resulting in a full dataset consisting of four horses for session 1, eight horses for session 2, and six horses (horses 1, 2, 5, 7, 10, and 12 from day 1) for session 3.

The average GPS/GNSS speed across all conditions was 16.48 ms^−1^ (first quartile (Q1): 15.86 ms^−1^; 3rd quartile (Q3): 17.10 ms^−1^). The average stride frequency was 2.47 Hz (Q1: 2.38 Hz, Q3: 2.55 Hz), and the average stride length was 6.69 m (Q1: 6.38 m, Q3: 6.98 m). (See Figure 1 for speed as a function of track hardness and moisture content).

### 3.3. Effect of Track Hardness and Moisture Content on Stride Frequency and Stride Length

At the average speed of 16.48 ms^−1^, the mixed model showed that shoeing condition had no significant effect on either stride frequency (*p =* 0.689) or stride length (*p =* 0.703) (Table 2 and Table 3). Both track hardness and moisture content, however, had significant effects on stride frequency and stride length (all *p* < 0.001; Table 2 and Table 3, Figure 2 and Figure 3).

For both stride frequency and stride length, mixed model analysis with speed as a covariate showed significant differences associated with track hardness and moisture content categories (Table 2 and Table 3). In both models, values measured on the hard surface were significantly different to the medium and soft track values (all pairwise comparisons including the hard track: *p* < 0.001 for both stride frequency and stride length). However, values acquired on the medium track were not different from the soft track (*p =* 0.168 for stride frequency and *p =* 0.515 for stride length). All pairwise comparisons between the different moisture levels were significantly different (all pairwise *p* < 0.001).

Due to the fact that only a limited number of track hardness and moisture conditions were present based on our categorizations of peak acceleration and percentage moisture content, the mixed model was not able to calculate a significance value for the two-way interaction between the two categories. In particular, a “hard” track category was only found on day 2 for the “dry” condition, and the “wet” condition was only associated with a “soft” track hardness category. However, it was possible to calculate estimated marginal means for both stride frequency and stride length for the existing hardness and moisture content combinations (Table 2 and Table 3). The smallest stride frequency value of 2.43 Hz was found on the soft track with a medium moisture level. The highest stride frequency value was found on the hard, dry track with a value of 2.50 Hz. The smallest stride length value of 6.60 m was found on the hard, dry track. The highest stride length was found on the soft track at medium moisture level at 6.78 m, a difference of 18.6 cm.

### 3.4. Effect of Session and Lane

For both stride frequency and stride length, mixed model analysis with speed as a covariate showed significant differences associated with session, lane, and their two-way interaction (Table 4 and Table 5). The majority of pairwise differences between session × lane combinations were significantly different; for exceptions, see Table 4 and Table 5. 

The lowest stride frequency value of 2.423 Hz was found in the middle lane of session 2 (soft track, medium moisture level, Table 1); the highest value of 2.540 Hz in the inside lane of session 3 (soft, wet track) closely followed by a value of 2.517 Hz on the outside lane of the same session (hard, dry) track. 

The lowest stride length value of 6.491 m was found for the inside lane of session 3, closely followed with a value of 6.553 m on the outside lane of the same session. The highest stride length value of 6.818 m was found on the middle lane of session 2 (soft track, medium moisture level), closely followed by the value of 6.779 m for the outside lane of the same session (medium hardness, dry track). The largest stride length difference between session × lane pairs was 32.7 cm, 76% higher than the largest difference between pairs of hardness × moisture combinations.

## 4. Discussion

In this study, simple track hardness and moisture level measurements in association with different dirt track preparations revealed changes in stride frequency and stride length. The change in stride length of 18.3 cm between the most different hardness × moisture combination, hard, dry track and soft, medium moisture level conditions, clearly exceeded the previously reported reduction in stride length of 10 cm over consecutive career race starts that has been associated with an increased injury risk (by a factor of 1.11) at racing speed in conjunction with an associated drop in speed [1]. Our approach was different, and we made use of the continuous speed outputs of the GPS/GNSS loggers for calculating stride frequency and stride length at a standardized speed. Nevertheless, taking into account track preparation, between races, as well as within a race (or within a training session), from start to finish or across the racetrack from inside to outside, together with precise location of each horse from the continuous GPS/GNSS data, may help with improving the precision of differentiating between alterations in stride parameters as a function of surface properties and changes prior to impending injuries. Track maintenance, for example, altering parameters such as cushion depth, using a harrowed or a sealed track, or variations in moisture contents, have been shown to influence the impact and loading characteristics of dirt tracks with vertical impact accelerations of a similar magnitude (19 to 66 × gravitational accelerations) [16] to those reported in our study (22 to 61 × gravitational acceleration) quantified with a tester designed for testing artificial turf surfaces [15].

Large stride length differences of up to 32.7 cm were found between sessions conducted on the two days, with modifications in track underlay compaction. This difference is larger than any difference between different track hardness and moisture level conditions. Moisture level variations appeared to be less consistently associated with changes in stride length or stride frequency (see Figure 2 and Figure 3). This suggests, that additional surface measurements might be required to better predict changes in stride parameters, including parameters such as impact firmness, cushioning, responsiveness, grip, and uniformity, measured previously [4]. It is promising that portable devices [3], in combination with moisture level measurements, have been shown to capture 80% of cushioning variability measurable with a gold standard surface tester [4]. This combination may provide a viable, more cost-effective alternative for delivering meaningful measurements that have previously been compared to subjective surface characteristics [17].

It is interesting that both the differences between hardness and moisture combinations, as well as between session and lane combinations, exceed 10 cm, the value that has been associated with an increased risk of injury by a factor of 1.11 over multiple races (career race starts) when measured specifically over the last furlong in each race [1]. Our statistical model has calculated stride frequency and stride length at the overall “standardized” average speed of 16.48 ms^−1^. This speed is considerably slower than what can be expected during racing, in which speeds up to and exceeding 20 ms^−1^ are not uncommon [9]. However, average speeds across 200 m sectionals toward the end of a race are around 16.6 ms^−1^, which is similar to the speed achieved by the horses in our study, and reduced speed and stride length are typically observed with race progression [9]. The comparatively large differences between surface preparations within the same surface suggests that, injury prediction models, whether ultimately applied to “in-race” data (for example, measured over the last furlong of each race [1]) or “in-training” data, might potentially benefit from inclusion of quantitative information about surface properties beyond surface type (turf, artificial, dirt) and categorical, subjective track condition, as determined by race stewards [1]. With GPS/GNSS, it is possible to quantify the location of each horse on the track both in terms of the distance from the start (along the track) and the distance from the inside rail. This might help with developing more detailed models expanding on the previous model using data from the last furlong of each race [1], taking into account surface property variations with progression over a race or training session or as a function of the horse utilizing more or less heavily frequented track areas, such as along the inside rail on a bend.

The higher speeds on the harder tracks during the third session appear to have been reached primarily through increased stride frequency, not increased stride length. On the harder track, the horses took a higher number of strides per second at the average speed of 16.48 ms^−1^; all EMM values for stride frequency for session 3 are higher than the stride frequency EMM values for sessions 1 and 2 (Table 4), while all stride length values on day 3 are smaller than those on day 1 (Table 5). The harder tracks on day 2 had been achieved through higher compaction and reduced harrowing depths. This had likely created complex changes beyond the peak acceleration (hardness) measured in this study, affecting firmness, cushioning, responsiveness, and grip of the surface. It might be possible to use portable equipment for measurements that have been linked to subjective assessments of equestrian surfaces [17]. The use of a mobile device might allow for a more widespread use and the collection of a larger database, thereby speeding up progress toward the goal of injury prediction.

With respect to track hardness, it is interesting to note that humans adjust their limb stiffness and maintain similar running mechanics, such as total vertical center of mass excursion and stride frequency [18]. That adjustment is achieved immediately (within the first step) after an abrupt surface change [19]. Horses, on the other hand, have limited capacity for adjusting their limb stiffness because the majority of leg length change is associated with the distal limb and the storage of energy in the flexor tendons with little influence of muscle activation [20]. In practice, horses therefore adjust their running mechanics and, for example, show increased vertical ranges of motion on softer ground in trot [21]. Consequently, it is not surprising to see changes in stride length and stride frequency in reaction to surface hardness in our present study.

Stride frequency increases with increasing speed between 0.009 Hz per ms^−1^ (session*lane model) and 0.018 Hz per ms^−1^ (hardness × moisture model). This represents a reduction in stride time of around up to 3 milliseconds for an increase in speed of 1 ms^−1^ at a stride frequency of 2.5 Hz. Stride length, on the other hand, increases by between 0.359 m per ms^−1^ for the hardness × moisture model and 0.386 m per ms^−1^ for the session*lane model, an increase of around 5% stride length at an average length of 6.6 m. Compared to the reported increase in stride frequency of 0.0305 Hz per ms^−1^ speed increase on an artificial surface [12], the horses in our study appear to show a smaller increase in stride frequency when speeding up. However, the previously reported increase in stride frequency was measured over a larger speed range from approximately 9 to 17 ms^−1^, while in our study, the jockey was asked to gallop at a given target speed of 16 ms^−1^, and as a result, speed varied comparatively little, with 50% of the recorded speed datapoints falling between 15.86 ms^−1^ and 17.10 ms^−1^, a variation of around +/−3.8% around the average speed. Further investigations should provide more data for the relationship between speed, stride frequency, and stride length on different surfaces and for different exercise conditions, including longitudinal effects with training [13] and running on curved tracks [14,22].

## 5. Conclusions

In relation to dirt racetrack hardness and moisture content, our study found that stride frequency was higher and stride length was reduced on the hard surface. Stride frequency and stride length showed more inconsistent changes with track moisture content. The lowest and highest stride frequencies were found on the soft track with the medium moisture level and on the hard, dry track, respectively. The lowest and highest stride lengths were found on the hard, dry track and the soft track at the medium moisture level, respectively. Stride length differences between these two categories were 18.6 cm, which is larger than a stride length reduction of 10 cm that was previously reported as leading to an increase in injury risk by a factor of 1.11 over consecutive career race starts [1].

There were larger differences in stride length (up to 32.7 cm) when investigating differences between data collection sessions compared to the highest stride length difference of 18.6 cm in relation to hardness and moisture content. This suggests that additional parameters might play a role when horses “select” a suitable gallop stride frequency and stride length combination in response to dirt racetrack preparation. This aspect needs further investigation.

## Figures and Tables

**Figure 1 sensors-24-02441-f001:**
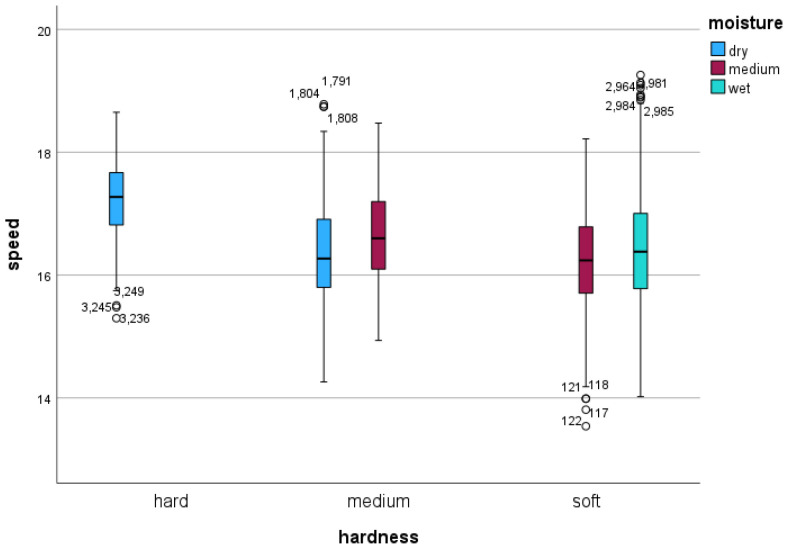
Speed (in ms^−1^) in association with track hardness category (*x*-axis categories) and track moisture content category (blue: dry condition; red: medium wet condition; green: wet condition). Speed measurements indicate that on the hard track, the horses show the highest speeds, while on soft and medium tracks horses gallop at lower speeds. Boxes represent 1st, 2nd (black horizontal line), and 3rd quartile. Whiskers extend to the highest and lowest values not considered outliers, with outliers defined as values that are more than 1.5 times the interquartile range below the first quartile or above the third quartile.

**Figure 2 sensors-24-02441-f002:**
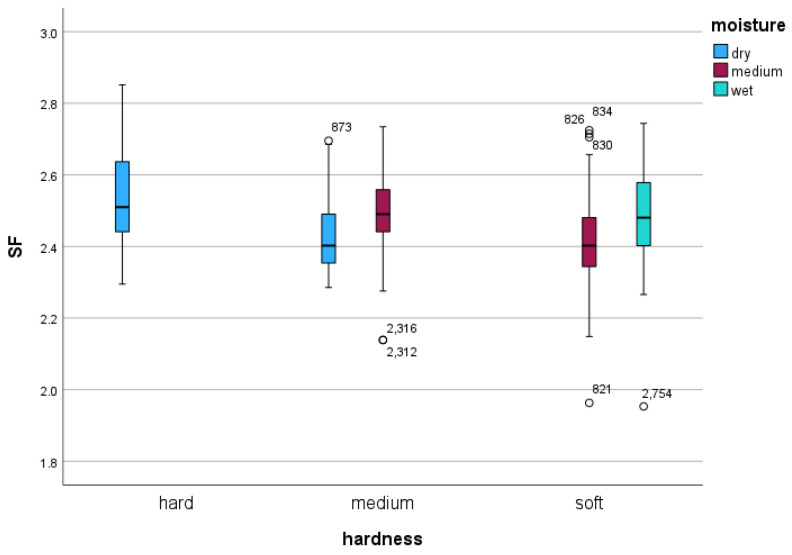
Stride frequency (in Hz) in association with track hardness category (*x*-axis categories) and track moisture content category (blue: dry condition; red: medium wet condition; green: wet condition). The plots illustrate an apparent interaction between hardness and moisture content with higher stride frequency values with increasing moisture content at a given hardness level. For definitions of boxes and whiskers see Figure 1.

**Figure 3 sensors-24-02441-f003:**
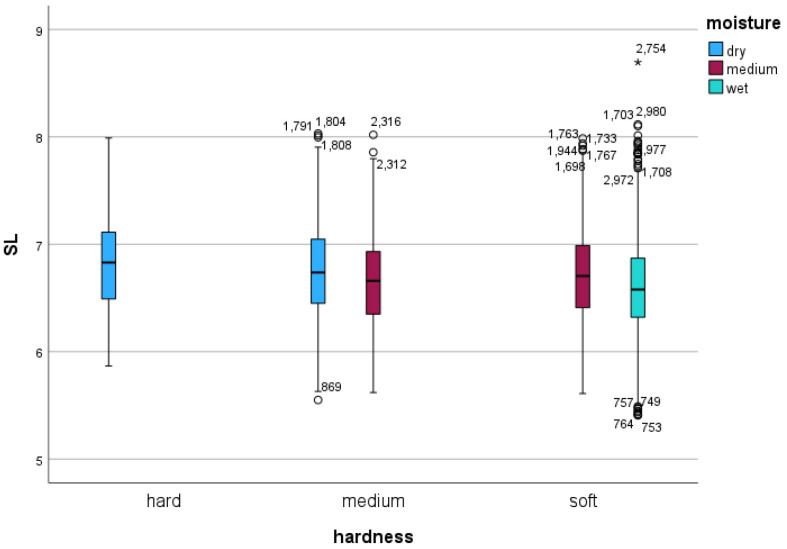
Stride length (in m) in association with track hardness category (*x*-axis categories) and track moisture content category (blue: dry condition; red: medium wet condition; green: wet condition). The plots indicate small differences in stride length within each hardness level with changes in moisture content. Note: Outliers in the plots are in the majority associated with the horse that galloped at the highest speed. For definitions of boxes and whiskers, see Figure 1.

**Table 1 sensors-24-02441-t001:** Track hardness (in multiples of gravitational acceleration) and moisture measurements (in %) on the two days. Day 1: morning and afternoon sessions (session 1 and session 2) with identical track preparation. Day 2: one session (session 3) with different (harder) track preparation.

	Hardness			Moisture	
	Mean	Category	Std	Mean	Category
session 1: day 1 morning					
inside	22.3 g	soft	4.4 g	23.3%	wet
middle	23.0 g	soft	2.1 g	17.8%	medium
outside	41.3 g	medium	9.2 g	12.6%	dry
session 2: day 1 afternoon					
inside	23.8 g	soft	3.6 g	27.4%	wet
middle	24.3 g	soft	0.8 g	18.6%	medium
outside	38.1 g	medium	7.7 g	13.4%	dry
session 3: day 2 morning					
inside	26.3 g	soft	4.7 g	21.8%	wet
middle	44.3 g	medium	4.2 g	18.3%	medium
outside	61.2 g	hard	14.8 g	12.2%	dry

**Table 2 sensors-24-02441-t002:** Effect of track hardness and moisture content on **stride frequency** in a mixed model analysis with speed as covariate. ^1^ Pairwise differences: conditions with same superscript not significantly different. **Note:** two-way interaction between hardness and moisture category could not be calculated because the “hard” track was only associated with a “dry” surface, and conversely, the “wet” surface was only associated with “soft” hardness category. EMM: estimated marginal mean values.

	*p*-Value	Covariate Estimate (95% Confidence Interval) or EMM (95% Confidence Interval)		
shoeing	0.689			
speed	**<0.001**	0.018 Hz/ms^−1^ (0.015 Hz/ms^−1^; 0.021 Hz/ms^−1^)		
hardness	**<0.001**	hard 2.502 (2.415; 2.589)	medium **^1^** 2.459 (2.372; 2.546)	soft **^1^** 2.464 (2.377; 2.551)
moisture	**<0.001**	dry 2.471 (2.384; 2.558)	Medium 2.456 (2.369; 2.543)	wet 2.493 (2.407; 2.580)
hardness × moisture	**NA**	hard × dry 2.502 (2.415; 2.589)	medium × dry 2.440 (2.353; 2.527) medium × medium 2.478 (2.391; 2.565)	soft × medium 2.434 (2.347; 2.521) soft × wet 2.493 (2.407; 2.580)

**Table 3 sensors-24-02441-t003:** Effect of track hardness and moisture content on **stride length** in a mixed model analysis with speed as covariate. ^1^ Pairwise differences: conditions with same superscript not significantly different. **Note:** two-way interaction between hardness and moisture category could not be calculated because the “hard” track was only associated with a “dry” surface, and conversely, the “wet” surface was only associated with “soft” hardness category.

	*p*-Value	Covariate Estimate (95% Conf. Interval) or EMM (95% Conf. Interval)		
shoeing	0.703			
speed	**<0.001**	0.359 m/ms^−1^ (0.351 m/ms^−1^; 0.367 m/ms^−1^)		
hardness	**<0.001**	hard 6.597 (6.393; 6.801)	medium **^1^** 6.712 (6.509; 6.916)	soft **^1^** 6.703 (6.500; 6.906)
moisture	**<0.001**	Dry 6.682 (6.479; 6.886)	medium 6.720 (6.516; 6.923)	wet 6.622 (6.419; 6.825)
hardness × moisture	**NA**	hard × dry 6.597 (6.393; 6.801)	medium × dry 6.768 (6.564; 6.971) medium × medium 6.656 (6.452; 6.860)	soft × medium 6.783 (6.580; 6.987) soft × wet 6.622 (6.419; 6.825)

**Table 4 sensors-24-02441-t004:** Effect of “session” and “lane” on **stride frequency** for mixed model analysis with speed as covariate. ^1^ Pairwise session differences: conditions with same superscript not significantly different.

	*p*-Value	Covariate Estimate (95% Conf. Interval) or EMM (95% Conf. Interval)		
Shoeing	0.660			
Speed	**<0.001**	0.009 Hz/ms^−1^ (0.005 Hz/ms^−1^; 0.012 Hz/ms^−1^)		
Session	**<0.001**	1: day 1 morning **^1^** 2.453 (2.370; 2.537)	2: day 1 afternoon **^1^** 2.446 (2.363; 2.530)	3: day 1 morning 2.515 (2.432; 2.598)
Lane	**<0.001**	inside 2.494 (2.411; 2.578)	middle 2.454 (2.371; 2.537)	outside 2.466 (2.383; 2.550)
session × lane	**<0.001**	session1 × inside 2.454 (2.380; 2.547) session1 × middle 2.451 (2.368; 2.535) session1 × outside 2.444 (2.361; 2.528)	session2 × inside 2.480 (2.396; 2.563) session2 × middle 2.423 (2.339; 2.508) session2 × outside 2.437 (2.354; 2.521)	session3 × inside 2.540 (2.456; 2.623) session3 × middle 2.488 (2.405; 2.572) session3 × outside 2.517 (2.434; 2.601)

All pairwise session × lane combinations are significantly different, except session1 × inside to session1 × middle (*p =* 0.779); session1 × inside to session2 × inside (0.510); session1 × middle to session1 × outside (*p =* 1.0); session1 × middle to session2 × outside (*p =* 0.933); and session2 × inside to session3 × middle (*p =* 1.0).

**Table 5 sensors-24-02441-t005:** Effect of “session” and “lane” on **stride length** for mixed model analysis with speed as covariate. ^1^ Pairwise session differences: conditions with same superscript not significantly different.

	*p*-Value	Covariate Estimate (95% Conf. Interval) or EMM (95% Conf. Interval)		
Shoeing	0.674			
Speed	**<0.001**	0.386 m/ms^−1^ (0.377 m/ms^−1^; 0.394 m/ms^−1^)		
Session	**<0.001**	1: day 1 morning **^1^** 6.728 (6.508; 6.949)	2: day 1 afternoon **^1^** 6.754 (6.533; 6.974)	3: day 1 morning 6.558 (6.338; 6.778)
Lane	**<0.001**	inside 6.620 (6.400; 6.840)	middle 6.726 (6.505; 6.946)	outside 6.695 (6.475; 6.915)
session × lane	**<0.001**	session1 × inside 6.703 (6.482; 6.924) session1 × middle 6.730 (6.509; 6.951) session1 × outside 6.752 (6.531; 6.972)	session2 × inside 6.664 (6.444; 6.885) session2 × middle 6.818 (6.598; 7.038) session2 × outside 6.779 (6.559; 6.999)	session3 × inside 6.491 (6.271; 6.712) session3 × middle 6.620 (6.408; 6.849) session3 × outside 6.553 (6.333; 6.774)

All pairwise session × lane combinations are significantly different, except session1 × inside to session1 × middle (*p =* 1.0); session1 × inside to session2 × inside (*p =* 1.0); session1 × middle to session1 × outside (*p =* 1.0); session1 × middle to session2 × outside (*p =* 0.218); session1 × outside to session2 × outside (*p =* 1.0); and session2 × middle to session3 × middle (*p =* 0.172).

## Data Availability

The dataset analyzed for this study is available on figshare (10.6084/m9.figshare.25395868).

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
