# Peer review of "Dirt Track Surface Preparation and Associated Differences in Speed, Stride Length, and Stride Frequency in Galloping Horses"

_sensors, 2024, doi:10.3390/s24082441_

Round 1

Reviewer 1 Report

Comments and Suggestions for Authors

This was a nicely presented manuscript. It was nice to set some metrics applied to track surface and impact on stride variables. 

I only had some minor comments

line 98 is it possible provide a reference to justify the mass and height for the testing protocol

Discussion

A major limitation with the Wong et al paper is the lack of quantitative data to describe track condition.  You have highlighted the need for quantitative data, is there a place somewhere in the discussion to indicate to the reader where in the normal distribution of dirt track surface where your conditions would sit ( i.e. within the normal variation experienced across a racing season, or at the upper and lower limits of what would normal be observed)?

line 326 lager speed --.....assume larger or greater 

Author Response

We thank the reviewer for their positive and constructive comments. Please see below our detailed responses as well as the original reviewer comments:

Reviewer 1

This was a nicely presented manuscript. It was nice to set some metrics applied to track surface and impact on stride variables. 

Thank you for your positive comments.

I only had some minor comments.

line 98 is it possible provide a reference to justify the mass and height for the testing protocol
We have added a reference to a previous study that has used this device. J. Wannop, S. Kowalchuk, M. Esposito, D. Stefanyshyn, Influence of Artificial Turf Surface Stiffness on Athlete Performance, Life 10 (2020) 340. https://doi.org/10.3390/life10120340.
We have also added more details about the testing standards that this device is adhering to: “ASTM testing standards F355 and F1936”

Discussion

A major limitation with the Wong et al paper is the lack of quantitative data to describe track condition.  You have highlighted the need for quantitative data, is there a place somewhere in the discussion to indicate to the reader where in the normal distribution of dirt track surface where your conditions would sit ( i.e. within the normal variation experienced across a racing season, or at the upper and lower limits of what would normal be observed)?

We currently have an ongoing study with weekly surface testing at two of the Alberta racetracks with a different (more mobile) testing device that can be easily used by the racetrack staff and can be carried around the track. So, unfortunately, we do not currently have good data to compare to. We have added a couple of sentences to the discussion referencing a previous publication that has investigated the influence of cushion depth and moisture content on harrowed and sealed horse dirt racetracks.
C.A. Mahaffey, M.L. Peterson, L. Roepstorff, The effects of varying cushion depth on dynamic loading in shallow sand thoroughbred horse dirt racetracks, Biosystems Engineering 114 (2013) 178–186. https://doi.org/10.1016/j.biosystemseng.2012.12.004.

line 326 lager speed --.....assume larger or greater 

changed

Reviewer 2 Report

Comments and Suggestions for Authors

I have read the manuscript titled "Dirt track surface preparation and associated differences in speed, stride length, and stride frequency in galloping horses" by Pfau et al. with interest. Congratulations on the manuscript. I have imagined how much hard work was involved, and it may have contributed to other researchers' training within the research group. Overall, from the reviewer's perspective, it is a well-planned and executed study with a simple and easy-to-understand formulation. However, a few points still need to be clarified before it is considered for publication.

Suggestions for improving the manuscript:

Title

My comments: The title is clear and informative;

Abstract:

My comments: The abstract is clear and informative

Introduction

Lines 37-41:  

My comments: The phrase needs to be shorter and more straightforward. Could you improve that?

Line 54: “There are also reported training effects”....

My comments: Which effects? Please, could specify it?

Line 66: every hypothesis must be written using the present tense verb.

My comments: Please change the future tense verb to present tense.

Materials and Methods

Lines 90-93: “After the first four horses... track underlay”.

My comments: This materials and methods part could have been more straightforward. Please consider making it more concise and clear.

Results

All tables and figures need to be improved for the manuscript. They need to be self-explanatory. 

Table 1:

My comments: Please, check the table formatting.

Figure 1:

My comments: Where are the figures 1A and 1B?

Discussion

The discussion is coherent and provides valuable information.

Line 320: typo

Conclusion

Your conclusion contains information about the results and some implications from your study, which must be included in the discussion. Please improve your conclusion by focusing on the relationship between the title and the study's goal. It needs to be more concise and less redundant.

Author Response

We thank the reviewer for their positive and constructive comments. Please see below our detailed responses as well as the original reviewer comments:

Reviewer 2

I have read the manuscript titled "Dirt track surface preparation and associated differences in speed, stride length, and stride frequency in galloping horses" by Pfau et al. with interest. Congratulations on the manuscript. I have imagined how much hard work was involved, and it may have contributed to other researchers' training within the research group. Overall, from the reviewer's perspective, it is a well-planned and executed study with a simple and easy-to-understand formulation. However, a few points still need to be clarified before it is considered for publication.

Suggestions for improving the manuscript:

Thank you for taking time to review our manuscript carefully. Please see our responses below and the changes in the manuscript.

Title

My comments: The title is clear and informative;

Thank you

Abstract:

My comments: The abstract is clear and informative

Thank you

Introduction

Lines 37-41:  

My comments: The phrase needs to be shorter and more straightforward. Could you improve that?

Thank you for highlighting this. We have broken this long statement up into multiple sentences and have rephrased accordingly.

Line 54: “There are also reported training effects”....

My comments: Which effects? Please, could specify it?

We have added some details here about changes in stride duration and duty factor.

Line 66: every hypothesis must be written using the present tense verb.

My comments: Please change the future tense verb to present tense.

Thank you for highlighting this. We have changed the hypothesis to present tense.

Materials and Methods

Lines 90-93: “After the first four horses... track underlay”.

My comments: This materials and methods part could have been more straightforward. Please consider making it more concise and clear.

We have rephrased to clarify the conditions for the three data collection sessions (session 1 through 3)

Results

All tables and figures need to be improved for the manuscript. They need to be self-explanatory. 

Table 1:

My comments: Please, check the table formatting.

We have reformatted table 1.

Figure 1:

My comments: Where are the figures 1A and 1B?

Thank you for highlighting this. We apologize for this mixup. Initially we had considered presenting two sub-panels but over the course of our internal revisions had decided against it. We obviously forgot to change the text that is accompanying each figure. We have revised this and apologize for this oversight.

Discussion

The discussion is coherent and provides valuable information.

Thank you for this positive comment.

Line 320: typo

Corrected

Conclusion

Your conclusion contains information about the results and some implications from your study, which must be included in the discussion. Please improve your conclusion by focusing on the relationship between the title and the study's goal. It needs to be more concise and less redundant.

Thank you. We have rephrased and shortened the conclusion section.

Reviewer 3 Report

Comments and Suggestions for Authors

Brief summary

This paper studies the relationship between track surface properties of hardness and moisture content with stride parameters of stride length and stride frequency. Studying this relationship is important as changes in stride length over time has been shown to be a predictor of injury. Importantly, this paper shows that more detailed measurements of track conditions might be useful for injury prediction models. 

Comments

General comments

Thank you for this paper. I have one major concern with the reference this paper makes to the Wong et al. 2022 paper. Caution needs to be taken as Wong’s paper is quoting that a decrease in stride length over time (career race starts) is predictive of injury and may not be the correct reference for the findings of this paper. Also, quoting that the difference in stride length of 0.186 m between hardness and moisture content being higher than the 0.10 m decrease being identified as an injury predictor in Wong’s paper is incorrect as Wong’s paper simply states that a decrease in stride length over time is a predictor of injury where the magnitude of decrease is inconsequential in this particular regard. For example, the 0.1 m decrease in stride length over time (career race starts) is associated with 1.18 times increase in risk of MSI and a 0.05 decrease in stride length would also be associated with an increased risk of MSI, just to a smaller extent. I believe the findings of this paper important but believe that the reference made to Wong’s paper highly incorrect. 

 Material and methods

Track preparation – (1) What was the decreasing depth of the surface cushion, was that measured? (2) Just a brief description of the track maintenance equipment used could be useful for the readers who are more interested/knowledgeable in the track surface and properties aspect of this paper  (3) How was the moisture content between the lanes (as well as the two days) varied? I don’t seem to see any information on how this was achieved. Was it simply a by-product of the cushion depth or was there a process in varying the moisture content?

Hardness and moisture measurements – Was there a basis for choosing the 9.1 kg mass being dropped from a height of 65.5 cm? A reference perhaps might be useful.

Data processing, track hardness – Was there a basis for the hardness categories assigned – was this assigned based on this experiment alone comparing the three lanes or referring to previous papers? References would be nice please.

Data processing, moisture content – Similar to above, any references to the moisture categories assigned?

Statistics – Am I right in assuming univariable models were not developed initially?  

Statistics , effect of hardness and moisture content – (1) I was initially a little confused with the usage of the terms “fixed covariate” and “fixed factors”. A suggestion for the sentences for lines 131 to 134 as follows: 

Mixed linear models for stride frequency and stride length as the outcome were implemented (SPSS v29, IBM, Armonk, NY, USA) with horse as random factor, and fixed factors including speed as a continuous covariate and surface characteristics (‘hardness’, ‘moisture’) and shoeing condition (‘aluminium’, ‘steel’, ‘barefoot’) and their two-way interactions as categorical covariates.

(2) Line 136 – Suggestion: To replace “final statistical model created” with “final multivariable model created”. 

Statistics,  effect of ‘session’ and ‘lane’: (1) Similar to above with the rewording of the sentence in lines 142 to 143 if the suggestion is taken. (2) Lines 146-148 - The sentence here should not be in the methods section. Consider deleting or moving elsewhere more suitable.

Results

Table 1 – outside session 1 std 9.2 % sign should be g

Figures 1, 2, and 3 – I’m not clear on what Figures 1(A) and 1(B), 2(A) and 2(B), 3(A) and 3(B) are referring to.

Figure 3 caption Line 185 – “..with the that galloped…” – missing word here?

Table 2 – Abbreviation EMM needs to be defined. If I’m not mistaken, it’s only mentioned for the first time in text in Line 216?

Discussion

Line 320 – Stride length - to capitalise the S if starting a new sentence was the intention?

Author Response

We thank the reviewer for their positive and constructive comments. Please see below our detailed responses as well as the original reviewer comments:

Reviewer 3:

Brief summary

This paper studies the relationship between track surface properties of hardness and moisture content with stride parameters of stride length and stride frequency. Studying this relationship is important as changes in stride length over time has been shown to be a predictor of injury. Importantly, this paper shows that more detailed measurements of track conditions might be useful for injury prediction models. 

Thank you for your summary.

Comments

General comments

Thank you for this paper. I have one major concern with the reference this paper makes to the Wong et al. 2022 paper. Caution needs to be taken as Wong’s paper is quoting that a decrease in stride length over time (career race starts) is predictive of injury and may not be the correct reference for the findings of this paper. Also, quoting that the difference in stride length of 0.186 m between hardness and moisture content being higher than the 0.10 m decrease being identified as an injury predictor in Wong’s paper is incorrect as Wong’s paper simply states that a decrease in stride length over time is a predictor of injury where the magnitude of decrease is inconsequential in this particular regard. For example, the 0.1 m decrease in stride length over time (career race starts) is associated with 1.18 times increase in risk of MSI and a 0.05 decrease in stride length would also be associated with an increased risk of MSI, just to a smaller extent. I believe the findings of this paper important but believe that the reference made to Wong’s paper highly incorrect. 

Thank you for highlighting that our manuscript was lacking specificity with respect to the comparison to the value of 0.1m from the previous injury prediction study. We have rephrased our references to that manuscript and now refer to the increased injury risk of 1.11 for each reduction in stride length of 10 cm (0.1m). With our approach and the use of continuous GPS/GNSS measurements we have used an approach that uses speed as a covariate and hence we calculate stride length and stride frequency at a ‘standardized’ speed, hence we cannot use the factor of 1.18 (referenced in the reviewer’s comment) referring to a decrease in speed by 0.1ms-1. We thank the reviewer for highlighting the shortcoming of our manuscript. We still believe that putting our measurements into the context of these changes measured prior to injury (as a function of career race starts) is a valid thing to do, since track properties will change between races and hence adding quantitative track measurements might be a helpful undertaking for finetuning longitudinal injury prediction models. We are now much more specific about the exact ‘value’ that we are referencing from Wong et al in our new version of the manuscript and highlighting that the published value is with reference to consecutive career race starts.

 Material and methods

Track preparation – (1) What was the decreasing depth of the surface cushion, was that measured?
This as not measured. The track preparation was undertaken by the racetrack staff who were asked to prepare the track as per usual conditions. No measurements, other than the ones outlines (impact accelerations, moisture content) were undertaken. We have clarified this in our updated manuscript and also now indicate that track preparation was undertaken so as to ‘mimic’ the conditions that would be used on a typical race day during the Calgary Stampede (on the first first experimental day in our study). On the second day, the aim was to achieve ‘faster’ conditions. This has been clarified in the materials and methods section of the updated manuscript.

(2) Just a brief description of the track maintenance equipment used could be useful for the readers who are more interested/knowledgeable in the track surface and properties aspect of this paper  
(3) How was the moisture content between the lanes (as well as the two days) varied? I don’t seem to see any information on how this was achieved. Was it simply a by-product of the cushion depth or was there a process in varying the moisture content?
(2) and (3): We have added more detail. The aim was to mimic the racing conditions encountered during the Calgary Stampede Chuckwagon races (on day1 of our experiments). On day2, the preparation crew was asked to prepare a ‘faster track’. No measurements of harrowing depths or the amount of water used were undertaken. We have specified this in the updated version of the materials and methods section.

Hardness and moisture measurements – Was there a basis for choosing the 9.1 kg mass being dropped from a height of 65.5 cm? A reference perhaps might be useful.
Thank you for highlighting this oversight. We have added a reference for the specific device used: J. Wannop, S. Kowalchuk, M. Esposito, D. Stefanyshyn, Influence of Artificial Turf Surface Stiffness on Athlete Performance, Life 10 (2020) 340. https://doi.org/10.3390/life10120340.
We have also added a couple of sentences to the discussion referencing a published study reporting impact accelerations in relation to cushion depth and moisture content on harrowed and sealed dirt horse racetracks to show that, while designed for ‘human athletes’, the testing device used here provides similar impact acceleration ranges to devices designed for horse racetrack testing:
C.A. Mahaffey, M.L. Peterson, L. Roepstorff, The effects of varying cushion depth on dynamic loading in shallow sand thoroughbred horse dirt racetracks, Biosystems Engineering 114 (2013) 178–186. https://doi.org/10.1016/j.biosystemseng.2012.12.004.

Data processing, track hardness – Was there a basis for the hardness categories assigned – was this assigned based on this experiment alone comparing the three lanes or referring to previous papers? References would be nice please.
This categorisation was based on our measurements and the values provided in our manuscript in Table 1. The specification of the device has been chosen according to ASTM testing standards F355 and F1936 and has been used for artificial turf testing in athletes. The ‘categorisation’ was chosen as a means of ‘illustrating’ the conditions encountered during ‘standard conditions’ (day 1) and conditions encountered when the track is prepared for ‘faster racing’ (day 2). This has been made clearer now in the materials and methods section.

Data processing, moisture content – Similar to above, any references to the moisture categories assigned?
This categorisation was based on our measurements and the values provided in our manuscript in Table 1. The ‘categorisation’ was chosen as a means of ‘illustrating’ the conditions encountered during ‘standard conditions’ (day 1) and conditions encountered when the track is prepared for ‘faster racing’ (day 2). This has been made clearer now in the materials and methods section. We have also added a reference (in the discussion) to a study that has reported moisture content (and cushioning depth) for horse dirt racetracks:
C.A. Mahaffey, M.L. Peterson, L. Roepstorff, The effects of varying cushion depth on dynamic loading in shallow sand thoroughbred horse dirt racetracks, Biosystems Engineering 114 (2013) 178–186. https://doi.org/10.1016/j.biosystemseng.2012.12.004.

Statistics – Am I right in assuming univariable models were not developed initially?  
This is correct, since stride length and stride frequency are highly dependent on speed, we believe it is highly important to include speed as a covariate.

Statistics, effect of hardness and moisture content – (1) I was initially a little confused with the usage of the terms “fixed covariate” and “fixed factors”. A suggestion for the sentences for lines 131 to 134 as follows: 

“Mixed linear models for stride frequency and stride length as the outcome were implemented (SPSS v29, IBM, Armonk, NY, USA) with horse as random factor, and fixed factors including speed as a continuous covariate and surface characteristics (‘hardness’, ‘moisture’) and shoeing condition (‘aluminium’, ‘steel’, ‘barefoot’) and their two-way interactions as categorical covariates.

We have rephrased similar to the above suggestion. We have split into two sentences.

(2) Line 136 – Suggestion: To replace “final statistical model created” with “final multivariable model created”. 
done

Statistics,  effect of ‘session’ and ‘lane’: (1) Similar to above with the rewording of the sentence in lines 142 to 143 if the suggestion is taken.
We have adapted accordingly.

(2) Lines 146-148 - The sentence here should not be in the methods section. Consider deleting or moving elsewhere more suitable.
Thank you for highlighting this. We agree that technically this would not go into the materials and methods section, however, in this case, we would like to keep the sentence in the materials and methods section to provide a clearer ‘justification’ why two different ‘sets of models’ were investigated which may not be clear to the uninitiated reader.

Results

Table 1 – outside session 1 std 9.2 % sign should be g
Thank you for spotting this!

Figures 1, 2, and 3 – I’m not clear on what Figures 1(A) and 1(B), 2(A) and 2(B), 3(A) and 3(B) are referring to.
We apologize about this. The text accompanying the figures has made references to non-existing sub-panels that we had used in our initial internal versions of the manuscript. Both panels had shown the same data but in different formats and over the course of our discussions we had decided to use the one shown in the manuscript. We have removed reference to the sub-panels that had been left in error in the originally submitted version of the manuscript.

Figure 3 caption Line 185 – “..with the that galloped…” – missing word here?
Indeed, ‘horse’ had been missing here. We have changed accordingly.

Table 2 – Abbreviation EMM needs to be defined. If I’m not mistaken, it’s only mentioned for the first time in text in Line 216?
Thank you. We have included a definition and have also used ‘confidence interval’ instead of ‘conf. interval’.

Discussion

Line 320 – Stride length - to capitalise the S if starting a new sentence was the intention?

Thank you. Indeed, a typo. Changed accordingly.